# Network Proximity-Based Drug Repurposing Strategy for Early and Late Stages of Primary Biliary Cholangitis

**DOI:** 10.3390/biomedicines10071694

**Published:** 2022-07-13

**Authors:** Endrit Shahini, Giuseppe Pasculli, Andrea Mastropietro, Paola Stolfi, Paolo Tieri, Davide Vergni, Raffaele Cozzolongo, Francesco Pesce, Gianluigi Giannelli

**Affiliations:** 1National Institute of Research IRCCS “Saverio De Bellis”, Castellana Grotte, 70013 Bari, Italy; raffaele.cozzolongo@irccsdebellis.it (R.C.); gianluigi.giannelli@irccsdebellis.it (G.G.); 2Department of Computer, Control and Management Engineering Antonio Ruberti (DIAG), Sapienza University of Rome, 00185 Rome, Italy; pasculli@diag.uniroma1.it (G.P.); mastropietro@diag.uniroma1.it (A.M.); 3National Research Council (CNR), Institute for Applied Computing (IAC), 00185 Rome, Italy; stolfi.p@gmail.com (P.S.); paolo.tieri@cnr.it (P.T.); davide.vergni@cnr.it (D.V.); 4Department of Emergency and Organ Transplantation, Nephrology, Dialysis and Transplantation Unit, University of Bari “A. Moro”, 70121 Bari, Italy; f.pesce81@gmail.com

**Keywords:** autoimmune liver disease, cholestatic diseases, primary biliary cirrhosis, primary sclerosing cholangitis, drug repurposing, network medicine

## Abstract

Primary biliary cholangitis (PBC) is a chronic, cholestatic, immune-mediated, and progressive liver disorder. Treatment to preventing the disease from advancing into later and irreversible stages is still an unmet clinical need. Accordingly, we set up a drug repurposing framework to find potential therapeutic agents targeting relevant pathways derived from an expanded pool of genes involved in different stages of PBC. Starting with updated human protein–protein interaction data and genes specifically involved in the early and late stages of PBC, a network medicine approach was used to provide a PBC “proximity” or “involvement” gene ranking using network diffusion algorithms and machine learning models. The top genes in the proximity ranking, when combined with the original PBC-related genes, resulted in a final dataset of the genes most involved in PBC disease. Finally, a drug repurposing strategy was implemented by mining and utilizing dedicated drug–gene interaction and druggable genome information knowledge bases (e.g., the DrugBank repository). We identified several potential drug candidates interacting with PBC pathways after performing an over-representation analysis on our initial 1121-seed gene list and the resulting disease-associated (algorithm-obtained) genes. The mechanism and potential therapeutic applications of such drugs were then thoroughly discussed, with a particular emphasis on different stages of PBC disease. We found that interleukin/EGFR/TNF-alpha inhibitors, branched-chain amino acids, geldanamycin, tauroursodeoxycholic acid, genistein, antioestrogens, curcumin, antineovascularisation agents, enzyme/protease inhibitors, and antirheumatic agents are promising drugs targeting distinct stages of PBC. We developed robust and transparent selection mechanisms for prioritizing already approved medicinal products or investigational products for repurposing based on recognized unmet medical needs in PBC, as well as solid preliminary data to achieve this goal.

## 1. Introduction

Primary biliary cholangitis (PBC) is a progressive immune-mediated liver disease that primarily affects women. The estimated incidence in Europe is about 1–2 per 100,000 population per year; commonly used ranges for incidence and prevalence per 100,000 are 0.3–5.8 and 1.9–40.2 [1], respectively.

PBC is characterised by chronic, cholestatic, granulomatous lesions and unusual destruction of intrahepatic bile ducts, as well as the presence of antimitochondrial antibodies. If left untreated, it leads to end-stage biliary cirrhosis. PBC has the highest concordance rate of all autoimmune diseases among identical twins, indicating a strong genetic predisposition [2]. Environmental factors also contribute to disease susceptibility [1]. The pathogenesis of the disease involves the interplay between immune and biliary pathways, which is fuelled by a cell-damage-dependent chronic process of cholestasis and liver fibrosis.

Additional advances in understanding the aetiology of hepatic injury in PBC have resulted from a deeper understanding of the gut–liver axis. In this setting, the farnesoid X receptor (FXR) is a critical transcriptional detector of bile acid metabolism, and one of its essential target genes in the gut is fibroblast growth factor (FGF)-19, which encodes an enterokine released into the portal blood and then binds to FXR [3].

Patients with PBC can remain clinically asymptomatic for a long time, possibly due to an early and slow-onset disease process [4]. Extrahepatic autoimmune diseases and hepatobiliary malignant tumours are typically caused by PBC [1]. Another important clinical aspect is that although late stages of PBC are temporarily responsive to liver transplantation, between 10.9 and 42.3 percent of patients develop recurrent PBC in the long term, resulting in graft loss and mortality [4]. Furthermore, as the pathogenesis of chronic fatigue appears to be commonly unrelated to the histological stage of PBC, as well as the degree of hepatic dysfunction or distinct serological markers of autoimmunity [4], it seems reasonable to take into account pathogenic factors outside the liver, with a key role of immune microenvironment, bone marrow microenvironment, and microbiota–gut mucosal interaction.

From a therapeutic standpoint, ursodeoxycholic acid (UDCA) is the first-line therapy for all PBC patients and has been shown to slow the disease progression [5,6]. Obeticholic acid is the sole second-line therapy for patients who do not respond to UDCA [7,8].

Despite recent advances in understanding of PBC pathogenesis, the management of the disease is very challenging. Because approximately 30 to 50 percent of patients with PBC have a partial biochemical response to UDCA, there is a need for alternative second-line intervention strategies [8].

Several drugs [9,10,11,12], including antifibrotic agents (colchicine, penicillamine, and malotilate), antivirals, silymarin, statins, thalidomide, and even targeted immunosuppressants, have been deemed partially useful, inefficient, or potentially dangerous in PBC in recent decades.

Drug repurposing (also known as drug repositioning, reprofiling, or retasking) is a method of discovering new applications for approved or investigational medications that go beyond the original medical indication [13]. This strategy has several advantages over developing a completely new drug for a specific indication. The risk of failure, for example, is reduced; because the repurposed drug has previously been demonstrated to be sufficiently safe in preclinical models and humans if early-stage trials have been completed, it is less likely to fail in later effectiveness trials, at least from a safety perspective. Furthermore, because the majority of preclinical testing, safety assessments, and, in some cases, formulation development will have already been completed, the time frame for drug development can be reduced.

Based on these premises, we reasoned that it would be highly relevant to establish a drug repurposing framework to find potential therapeutic agents in PBC and based our study on the identification of drugs targeting relevant pathways derived from an expanded pool of genes involved in different stages of the disease. To that end, we exploited the Scheuer histological staging system [14].

## 2. Materials and Methods 

This study was carried out via the following steps: (i) a comprehensive collection of data existing in the literature was carried out to compile a list of genes associated with the disease; (ii) a network medicine approach was applied to filter and prioritize the collected data; (iii) functional analysis of the genetic information was performed to provide the basis for the drug repurposing report. Details are provided below.

### 2.1. Disease-Associated Gene Data Gathering

A thorough search and filtering of the literature and databases were performed to compile a comprehensive genetic landscape of PBC. From 1990 to October 2021, we interrogated available electronic repositories (MEDLINE, PubMed, DisGeNET [15]) using the MeSH terms (“primary biliary cholangitis” OR “primary biliary cirrhosis” OR “PBC”) AND (“genes” OR “gene”) to gather genes associated with the disease (DAGs, i.e., disease-associated genes or “seed genes”). Such genes were also labelled by disease stage (early or late PBC) when clinically feasible or as “unspecified stages” (US) otherwise. This search included human studies, and non-English articles were excluded from the analysis.

Studies with non-retrievable online data (for example, genes related to PBC and disease stages), missing content, or unclear information were excluded from the analyses. All the latter processes were conducted in R environment version 4.0.5 (31 March 2021).

Figure 1 shows the PRISMA flow diagram for study inclusion. The identification of features that characterize DAGs, namely genes experimentally associated with a specific disease, is critical for determining a complete genetic description of the pathology, assisting in the discovery of its aetiology and potential treatments. Given a starting set of “seed” genes, the presence of characteristic patterns in their genetic, functional, or topological features can be used to better understand the disease’s characteristics and to uncover other associated genes.

Each gene is assigned a “relevance score” that represents the degree of certainty that a seed gene is relevant to the disease. This score is assigned a value equal to the DisGeNET gene–disease association (GDA) score for those genes present in the DisGeNET data source. The GDA score ranges from 0 to 1 and is computed using the number and type of sources (level of curation and model organisms) and the number of publications supporting the association [15]. 

The latter score is computed as:
GDA score=C+M+I+L
where 
C
 depends on the number of curated sources that support the GDA, 
M
 depends on the supporting non-human sources, 
I
 relies on sources from GWAS studies et similia, and 
L
 is the number of publications from text mining sources supporting the GDA.

With regard to the remaining genes (manually curated from PubMed and MEDLINE data sources), we assigned as the maximum GDA score of the corresponding PBC disease stage (i.e., early stages, late stages, unspecified stages). This choice was made in order to assign a higher weight to manually curated genes, which, thanks to the specific selection process, can be associated with the disease more reliably and robustly. Such genes have a major impact on the network diffusion process.

### 2.2. Network-Based Putative Disease Gene Prioritization

Following the identification of the set of DAGs, we collected updated human protein–protein interaction (PPI) data from the curated biomedical interaction repository BioGRID [16] as the first step to proceed with prioritisation (i.e., a rank of their importance in the disease) of the DAGs via a network medicine approach. Such a computational approach exploits topological (i.e., related to the network structure) information deriving from the reconstruction and the analysis of protein interaction networks, as well as other features, to provide insights about the role of the gene in the onset and the development of the disease [17,18]. Specifically, we applied a combined approach consisting of a network diffusion process and machine learning (ML) models [19] to provide a PBC “proximity” or “involvement” gene ranking. 

Network propagation (or network diffusion or heat diffusion) is a valuable and extensively employed process in network medicine and systems biology with relevance in disease gene prioritisation, drug repurposing, and patient stratification, among other applications [20]. It consists of the exploitation of the concept of heat diffusion; like a flow of heat diffuses over time in a medium, in a PPI network, a given amount of “virtual” heat flows from nodes where it is higher toward nodes where it is lower according to their mutual connections (and, optionally, to other features). In practice, the process is accomplished starting with a subset of “hot” seed nodes (here, the DAGs), after which such “heat” diffuses through the network according to its topology. After an arbitrary time-lapse, the final heat distribution (which generally favours nodes proximal to the “hot” nodes) is quantified and generates a proximity ranking that can be used to identify a subset of genes that are closely associated with the selected seed genes.

The rankings from the diffusion process are then exploited via an adaptive positive-unlabelled (APU) machine learning model to provide sets of reliably putative disease-associated genes (“likely positive” (LP) genes). The APU ML model is an ML setting in which only a set of positive instances is labelled (i.e., “true” seed genes), whereas the rest of the dataset is unlabelled, overcoming the issue of sharp binary classification (true/false or “associated to the disease”/“not associated to the disease”). The unlabelled instances may be either unspecified positive genes (the predicted putative genes) or true negative genes (not associated with the disease). At the end of this step, a set of putative disease-associated LP genes is yielded.

The top LP genes were added to the original PBC-related genes (validated by prior studies) to provide the final dataset of in silico genes most likely associated with PBC. The detailed mathematical and algorithmic procedure is described in Appendix A.

### 2.3. Functional Annotation

WebGestalt (WEB-based Gene SeT Analysis Toolkit) analysis was performed as previously reported. [21] The following parameters were used in the pathways and drug repurposing enrichment analysis. Homo sapiens was chosen as the model organism (human). KEGG, Reactome, DrugBank, and GLAD4U resources were used as enrichment categories. The human genome was used as a reference list for all mapped gene symbol IDs from the specified platform, utilising all genome protein-coding genes as background genes. In each category, the minimum number of IDs was 5, and the maximum number of IDs was 2000. Next, Fisher’s exact test-based over-representation enrichment analysis (ORA) was conducted. The Benjamini–Hochberg method was used to account for a significant false discovery rate (FDR) of less than 0.05.

Figure 2, Figure 3 and Figure 4 depict the analysis workflow from the initial curated genes (from the literature review and the DisGeNET database) to the LP genes obtained from the application of the proposed network-based putative disease gene prioritization algorithm.

## 3. Results

### 3.1. Identification of Seed Genes

Study selection: As shown in Figure 1, all articles used in this document were screened for eligibility based on their titles and abstracts. Then, the full text of all selected studies was thoroughly reviewed. Two investigators (E.S. and G.P.) handled data based on predefined eligibility criteria, and a third (F.P.) investigator resolved inconsistencies after discussion. The two review authors extracted data on PBC-related genes independently and summarised these gene characteristics in a table (Appendix A). A total of 1498 curated seed genes were found, which were then reduced to 1121 after data cleansing (duplicated removal after correcting gene symbol names to the HGNC human gene symbol standard [23]). These 1121 seed genes were then labelled according to their PBC stage as n = 238 (early stages), n = 183 (late stages), or n = 728 (US), depending on the information provided by the specific article.

### 3.2. Over-Representation Analysis (ORA)

Using the novel propagation algorithm described in Section 2 (and Appendix A), 150 LP genes, each with their relevance score, were identified as potential candidates for drug repurposing and pathway analysis from each of the PBC stages considered, in addition to the original seed genes used as inputs for the modified APU algorithm (early stages, late stages, and US). Appendix A show the results for each of these stages, as well as the input data (either original seed genes or LP genes obtained from our algorithm) obtained from the ORA in WebGestalt. The presence of previous or ongoing clinical trials was checked on https://clinicaltrials.gov/ (accessed on 23 February 2022)

### 3.3. Drug Repurposing

#### 3.3.1. Seed Gene Drug Repurposing ORA

With regard to the seed genes, gene-drug predicted analysis targeted a variety of existing drugs. In particular, regardless of the PBC stage, the most enriched and significant finding was taurocholic acid (TUDCA) (*p* = 3.77 × 10^−15^, FDR = 1.33 × 10^−13^), followed by anakinra (*p* = 1.98 × 10^−13^, FDR = 5.96 × 10^−13^). In particular, TUDCA was repurposed by 27 genes.

Moreover, another significant and enriched result was antivirals for systemic use (*p* < 3.33 × 10^−16^, FDR < 1.25 × 10^−14^) by 28 genes, specific antirheumatic agents (*p* < 3.33 × 10^−16^, FDR < 1.25 × 10^−14^) by 15 genes, etanercept (*p* = 1.62 × 10^−9^, FDR = 2.96 × 10^−8^) by 13 genes, mycophenolate mofetil (*p* = 9.42 × 10^−7^, FDR = 8.68 × 10^−6^) by 9 genes, interleukin inhibitors (*p* < 3.33 × 10^–16^, FDR < 1.25 × 10^–14^) by 74 genes (i.e., CCL2, CCL20, CCL26, CCL27, CD28, CD40, CD69, CXCL3, CXCL8, IFNG, IL10, IL12A, IL12B, IL13, IL16, JAK2, NOS2, IL17A, IL1A, IL1B, and many other interleukin/receptor factors, as well as matrix metallopeptidase, innate immune signal transducers, and Toll-like receptors), TNF-alpha inhibitors (*p* < 3.33 × 10^−16^, FDR < 1.25 × 10^−14^) by 47 genes (i.e., CCL2, CCL20, CD40, CHUK, CXCL8, HMGB1, IFNG, VCAM1, TNF receptors, interleukin/receptor factors, matrix metallopeptidase, and Toll-like receptors), drugs for musculoskeletal system disorders (*p* = 3.33 × 10^−16^, FDR = 1.25 × 10^−14^) by 28 genes, corticosteroids, potent (group III) (*p* = 3.22 × 10^−4^, FDR = 1.57 × 10^−3^) by 6 genes, specific immunoglobulins (*p* < 3.33 × 10^−16^, FDR < 1.25 × 10^−14^) by 57 genes (i.e., AIRE, AQP4, ARID3A, BANK1, CD14, CD19, CD1D, CD274, CD28, CD40, CD40LG, CD79A, COL17A1, CTAG1B, CTLA4, CXCR5, FYN, ICOS, IFNG, IGHG3, and many interleukins, as well as integrin, cell adhesion molecules, innate immune signal transduction adaptor, and protein tyrosine phosphatase), monoclonal antibodies (*p* < 3.33 × 10^−16^, FDR < 1.25 × 10^−14^) by 59 genes (immune-regulatory receptors, cell adhesion molecules, integrin, and interleukins, such as BCAP31, CCR5, CD14, CD19, CD1D, CD226, CD244, CD274, CD28, CD40, CD40LG, CD48, CD69, CD72, CD74, CD80, CD86, CD96, CD99, COL17A1, CTLA4, FASLG, ICOS, ICOSLG, ITGA1, ITGA5, ITGAL, ITGAV, JAM3, KLRG1, KRT20, L1CAM, LAMA4, LILRB2, LILRB3, LTB, LTBR, MS4A1, PDCD1, PDCD1LG2, PECAM1, PIK3CA, PLAUR, POLA1, SLAMF6, VCAM1, VEGFA, VTCN1, several interleukins, and TNF subfamily receptors).

Additional results included antigout preparations (*p* = 1.57 × 10^−4^, FDR = 8.61 × 10^−4^) repurposed by 9 genes, antineovascularisation agents (*p* = 3.33 × 10^−16^, FDR = 1.25 × 10^−14^) by 47 genes (i.e., ADRB2, APEX1, ATM, BCR, CCR5, CD274, CDKN1B, CXCL8, CYP2E1, CYP3A4, DEFB1, DEFB4A, FOXO3, HAMP, HIF1A, HMGB1, HMOX1, LTF, NFKB1, NR1I2, PDCD1, PDGFRA, TPMT, VCAM1, PIK3CA, PPARA, SLC22A1, VEGFA, VEGFB, and many ATP-binding transporters, as well as solute carriers, signal transducers, and activators of transcription), biguanides (*p* = 4.60 × 10^−4^, FDR = 2.15 × 10^−3^) by 9 genes, simvastatin (*p* = 1.41 × 10^−5^, FDR = 1.02 × 10^−4^) by 14 genes, doxorubicin (*p* = 3.56 × 10^−6^, FDR = 2.95 × 10^−5^) by 20 genes, tamoxifen (*p* = 3.14 × 10^−4^, FDR = 1.56 × 10^−3^) by 12 genes, protein kinase inhibitors (PKIs) (*p* = 3.33 × 10^−16^, FDR = 1.26 × 10^−14^) by 61 genes (i.e., ABCG2, AREG, ARRB1, BCR, CCN2, CHRM3, CHUK, CSF1R, CXCL8, DDR1, DDR2, DGKQ, ETS1, FOXO1, FOXO3, FYN, HGF, HIF1A, HMOX1, IRAK1, ITGAV, JAK2, KLF4, MAP3K14, MMP13, MYD88, NFKB1, NTRK2, PDE5A, PDGFRA, PDK4, PIK3CA, PIM2, PIN1, PRKCB, PSMD9, PTGS2, PSMD9, PTGS2, RIPK3, RPS6KA4, RPS6KB1, SLC22A1, SLC22A1, SMAD2, SMAD3, TAB1, TAB1, TGFB1, TNFSF10, TRAF3, TYK2, many cyclin-dependent kinases, suppressors of cytokine signalling/signal transducers, chemokine ligand, and protein tyrosine kinase), enzyme inhibitors (*p* = 3.48 × 10^−14^, FDR = 1.10 × 10^−12^) by 62 genes (i.e., ACE, ACE2, AGT, ATM, BCR, CDK1, CDKN2A, CHUK, CSNK2A2, CST7, CTSL, CXCL8, CYP3A4 DNMT1, FOXO1, FOXO3, GZMB, HDAC9 HGF, HIF1A, HMOX1, ITIH4 JAK2, KLF4, NFKB1, NOS3, NFKB1, PDE5A, PDGFRA, PIK3CA, PIM2, PLAUR, PRKCB, SPINT1, TGM2, TNFSF10, TPMT, VCAM1, and ATP-binding transporters, as well as matrix metallopeptidase, cyclin-dependent kinase, nuclear receptors, serpin subfamily, solute carriers, and nuclear receptors), EGFR inhibitors (*p* = 8.53 × 10^−3^, FDR = 2.54 × 10^−2^) by 8 genes, and macrolides (*p* = 1.52 × 10^−3^, FDR = 5.99 × 10^−3^) by 13 genes.

Abciximab was the most enriched drug for the seed genes classified as early stages (*p* = 1.13 × 10^−5^, FDR = 8.55 × 10^−3^), followed by muromonab (*p* = 1.54 × 10^−5^, FDR = 9.25 × 10^−3^) and artenimol (*p* = 3.08 × 10^−5^, FDR = 1.3 × 10^−2^). Artenimol was repurposed by seven genes (i.e., CCT3, LGALS1, LGALS1, MAP4, RPL10, RPS13, RPS28, and ZYX), muromonab by four genes (C1R, C1S, FCGR2b, and VTN), and abciximab by four genes (C1R, C1S, CD3D, and FCGR2B).

Seed genes classified as late stages found epipodophyllotoxin (*p* = 1.94 × 10^−10^, FDR = 1.17 × 10^−7^) and TNF-alpha (*p* = 2.26 × 10^−9^, FDR = 8.51 × 10^−7^) as the most enriched drugs. ABCB1, BAX, BCL2, FADD, FAS, and several tumour protein families were identified as the genes responsible for such findings.

#### 3.3.2. In Silico Gene Drug Repurposing ORA

The same type of analysis was performed on the LP genes (propagated from seed genes) obtained using our novel network propagation algorithm.

The most enriched drugs found in LP genes obtained from seed genes with no details of their PBC disease stage were geldanamycin (*p* < 3.33 × 10^−16^, FDR < 1.25 × 10^−14^), staurosporine (*p* = 1.33 × 10^−^^15^, FDR = 4.45 × 10^−^^13^), and PKI (*p* < 3.33 × 10^−16^, FDR < 1.25 × 10^−14^). The significance of geldanamycin was attributed to 16 genes in particular (AKT1, BCL, EGFR, ERBB2, UBB, and many members of the heat shock protein family). Staurosporine was repurposed by 17 genes (AKT1, APP, BIRC3, CDC37, ELAVL1, GSK3B, and many members of the heat shock protein family), and PKIs were repurposed by 53 genes (i.e., AKT1, BIRC, CAV1, CBL, CDC37, CDC42, CDK2, COPS5, CRKL, and CSK).

PKIs were also found to be enriched in LP genes derived from PBC seed genes from early stages, as well as antineovascularisation agents (*p* = 9.3481 × 10^−^^14^, FDR = 5.6294 × 10^−^^11^; 22 genes involved: APEX1, AURKA, CDK2, EGFR, PIK3CA, TNF, and other members of the heat shock protein family). More intriguing was the branched-chain amino acids (BCAAs) (i.e., l-lysine, l-threonine, both with significant p and FDR values of 0) and enzyme inhibitors (*p* = 5.78 × 10^−^^11^, FDR = 1.74 × 10^−^^8^), which were discovered to be clinically acceptable for such early disease stages.

With regard to LP genes obtained from PBC seed genes from late stages, erlotinib (*p* = 3.24 × 10^−^^14^, FDR = 2.32 × 10^−^^11^), geldanamycin (*p* = 1.38 × 10^−^^12^, FDR = 4.63 × 10^−^^10^), and EGFR inhibitors (*p* = 3.43 × 10^−^^13^, FDR = 1.46 × 10^−^^10^) were the most enriched drugs. In particular, erlotinib was found to be repurposed by 11 LP genes (YWHAQ, STUB1, STAT3, KRAS, and PTEN, among the others).

### 3.4. Pathway Analysis

#### 3.4.1. Seed Gene Pathway ORA

In terms of the seed genes, the gene-pathways predicted analysis accurately targeted a wide range of existing pathways, regardless of PBC stage. The most enriched and significant discovery was the immune system (*p* < 3.33 × 10^−16^, FDR < 1.25 × 10^−14^) repurposed by 203 genes, cytokine signalling in the immune system (*p* < 3.33 × 10^−16^, FDR < 1.25 × 10^−14^) by 114 genes, pathways in cancer (*p* < 3.33 × 10^−16^, FDR < 1.25 × 10^−14^) by 77 genes, signalling by interleukins (*p* < 3.33 × 10^−16^, FDR < 1.25 × 10^−14^) by 84 genes, cytokine–cytokine receptor interaction (*p* < 3.33 × 10^−16^, FDR < 1.25 × 10^−14^) by 73 genes, measles (*p* < 3.33 × 10^−16^, FDR < 1.25 × 10^−14^) by 34 genes, Th17 cell differentiation (*p* < 3.33 × 10^−16^, FDR < 1.25 × 10^−14^), by 30 genes, the Toll-like receptor signalling pathway (*p* < 3.33 × 10^−16^, FDR < 1.25 × 10^−14^) by 31 genes, the NF-kappa B signalling pathway (*p* < 3.33 × 10^−16^, FDR < 1.25 × 10^−14^) by 32 genes, and inflammatory bowel disease (IBD) (*p* < 3.33 × 10^−16^, FDR < 1.25 × 10^−14^) by 31 genes.

The most enriched and significant findings from PBC early stages was innate immune system (*p* = 4.6435 × 10^−12^, FDR = 1.3254 × 10^−8^) by 37 genes, Toll-like receptor cascades (*p* = 3.36 × 10^−9^, FDR = 3.03 × 10^−6^) by 13 genes, and complement and coagulation cascades (*p* = 4.25 × 10^−9^, FDR = 3.03 × 10^−6^) by 10 genes.

The most enriched and significant discoveries from late stages, on the other hand, monoclonal antibodies (*p* < 3.33 × 10^−16^, FDR < 1.25 × 10^−14^) by 26 genes, Epstein–Barr virus infection (*p* = 6.66 × 10^−16^, FDR = 7.60 × 10^−13^) by 20 genes, pathways in cancer (*p* = 1.22 × 10^−15^, FDR = 1.16 × 10^−12^) by 29 genes, immunoglobulins (*p* = 7.99 × 10^−15^, FDR = 5.70 × 10^−12^) by 32 genes, and the IL-18 signalling pathway (*p* = 3.89 × 10^−13^, FDR = 2.46 × 10^−10^) by 20 genes.

#### 3.4.2. In Silico Gene Pathway ORA

With regard to LP genes, regardless of their PBC disease stage, the B-cell receptor signalling pathway was the most enriched (*p* < 3.33 × 10^−16^, FDR < 1.25 × 10^−14^), enriched by 16 genes (AKT1, mitogen-activated proteins, and inhibitors of nuclear factor-kappa B-kinase subunits), followed by other interesting pathways, such as the neurotrophin signalling pathway (*p* < 3.33 × 10^−16^, FDR < 1.25 × 10^−14^), enriched by 19 genes (CALM2/3, mitogen-activated proteins, TP53, RELA, and others).

Interestingly, LP genes propagated from initial PBC early stages seed genes were found to be enriched for ubiquitin-mediated proteolysis pathways (*p* = 6.84 × 10^−^^14^, FDR = 3.07 × 10^−^^11^) by 17 genes (BIRC3, genes from the cullin family, FBX11, FBXW7, VHL, and WWP2 genes) and integrated breast cancer pathway (*p* = 3.88 × 10^−^^14^, FDR = 2.09 × 10^−^^11^) by 18 genes (AR, BRCA1, MYC, KRAS, EGFR, and USP15, among others).

Finally, LP genes propagated from initial PBC late stages seed genes were found to be enriched for complement and coagulation cascades (*p* = 9.78 × 10^−^^9^, FDR = 8.98 × 10^−^^7^) by 15 genes (C1QA, CFH, F3, SERPING1, and other complements) and allograft rejection pathway (*p* = 9.99 × 10^−^^5^, FDR = 2.03 × 10^−^^3^) by 7 overlapping genes (LOC102723407, HLA-DRA, and other major histocompatibility complexes).

## 4. Discussion

We aimed to investigate new potential disease genes involved in different stages of PBC using a robust base of previously validated PBC disease genes and a novel propagation algorithm. We reasoned that developing a drug repurposing framework would be extremely useful in identifying potential therapeutic agents in PBC, so we identified drugs targeting relevant pathways to aid in the understanding of the PBC molecular landscape, as well as the identification of genes that are not directly associated with it.

Figure 5 portrays the top drugs and pathways based on our findings. Concerning the interaction with seed genes, we confirmed that interleukin/protein kinase/TNF-alpha inhibitors, drugs for musculoskeletal system disorders, TUDCA, immunosuppressants, antirheumatic agents, and simvastatin and atorvastatin were among the potential drug classes, regardless of PBC stage. Additionally, the immune system, cancer pathways, interleukin signalling, cytokine–cytokine receptor interaction, Th17 cell differentiation, and Toll-like receptor (TLR)/NF-kB signalling/IBD pathways were the most involved in PBC.

A substantial body of evidence suggests that NF-kB signalling plays a role in immunity, inflammation, cancerogenesis, and nervous system function [24]. Indeed, PBC has been shown to activate the TLR4/MyD88/NF-kB signalling pathway in mice, causing the release of inflammatory molecules, as well as the production of a significant number of apoptotic proteins, resulting in hepatocellular injury [25]. The bacterial lipopolysaccharide receptor TLR4 was found to be overexpressed in vascular endothelial/bile duct cells and periportal hepatocytes, implying that bacterial pathogens and TLR4 may be involved in the inflammatory processes of PBC livers.

Ustekinumab, a p40 subunit antagonist of interleukin-12 and interleukin-23, is used to treat patients with moderate-to-severe ulcerative colitis or moderate-to-severe active Crohn’s disease [26]. In UDCA-unresponsive PBC adults, ustekinumab therapy was associated with modest benefits [27]. Anakinra, a recombinant IL-1 receptor antagonist, was effective in the treatment of patients with severe bacterial sepsis. However, no clinical trials on PBC are currently underway [28].

In a short-term trial, a UDCA-unresponsive PBC patient displayed a dramatic response to the PKI baricitinib, despite the existing risk of developing meaningful adverse events [29].

Concerning the potential of UDCA and TNF-alpha inhibitors, the relationship between serum TNF-alpha and TGF-beta levels and PBC severity has been established, reflecting disease severity, as have the effects of UDCA medication on lowering mentioned cytokine levels in late stages. Our findings confirmed the already known efficacy of TUDCA in PBC, which also exerts FXR-agonist activity [30,31,32].

Drugs that may modulate immunological abnormalities in PBC have been investigated, and budesonide, ciclosporin, and rituximab have all shown promise in slowing disease progression [33]. Leflunomide is an isoxazole-derivative antirheumatic drug that works by inhibiting pyrimidine synthesis. It has been used successfully as a viable alternative to methotrexate in the treatment of rheumatoid arthritis [34]; however, there are currently no clinical trials on PBC.

In terms of statins, only a few studies on PBC have been conducted. In particular, atorvastatin and simvastatin did not improve PBC-related cholestasis, especially in patients with an incomplete biochemical response to UDCA [35,36].

Concerning drugs for musculoskeletal system disorders [37], it has been suggested that the RANK-RANKL axis may influence PBC beyond the osteoclastogenesis effects. A study found that PBC patients had significantly higher levels of RANK in cholangiocytes and RANKL in CD4, CD8, and CD19 cells surrounding bile ducts than those with other liver diseases, implying that the RANK-RANKL axis plays a role in the process of bile duct injury. Denosumab, a RANKL human monoclonal inhibitor primarily used to treat osteoporosis, has been shown to preserve bone metabolism/liver function and has never been investigated in PBC.

From PBC early stages, the most enriched and significant findings for seed genes were the innate immune system, Toll-like receptor cascades, complement, and coagulation cascades. Monoclonal antibodies, Epstein–Barr virus infection, cancer pathways, immunoglobulins, and the IL-18 signalling pathway were the most enriched and significant discoveries from late stages. Additionally, according to our enrichment analysis, abciximab, muromonab, and artenimol were the most likely candidates for early PBC stages, whereas epipodophyllotoxin was a possibility for late stages.

Abciximab, a glycoprotein IIb/IIIa inhibitor, has been used in the past to reduce myocardial ischemic complications due to its anti-inflammatory properties [38].

The murine anti-CD3 monoclonal antibody muromonab (OKT3) effectively blocks all human T-cell functions. Prophylaxis with muromonab as an induction therapy, together with corticosteroids, azathioprine, and postponed cyclosporin, optimizes early graft function following organ solid transplantation by delaying the adverse events of cyclosporin until graft function is established [39,40].

Additionally, artenimol (artemisinin derivative) is used as an antimalarial agent. Artemisinins are thought to bind to haem within the infected erythrocytes [41].

Epipodophyllotoxins derivatives are currently used in cancer therapy. Etoposide and teniposide are two examples [42]. They are anticancer drugs that work by inhibiting topoisomerase II, which has excellent activity against both drug-sensitive and drug-resistant cancer cells. Etoposide was recently reformulated for the treatment of cytokine storms in COVID-19 patients [43].

When we looked at the LP genes, we discovered that antineovascularization agents, BCAAs, and enzyme inhibitors were the most promising agents for the early stages, whereas EGFR inhibitors, such as erlotinib, were only effective in the late stages and geldanamycin and staurosporine were potentially effective for US of PBC.

Indeed, plasma BCAA patterns in patients with PBC and primary sclerosing cholangitis (PSC) are markedly abnormal [44]. In particular, diminished levels of these amino acids (particularly l-phenylalanine and l-tyrosine) have been linked to chronic fatigue, particularly in PBC [45]. The mammary gland development pathway—involution (stage 4 of 4)—the proven master regulator of which is signal transducer and activator of transcription 3 (STAT3) [46,47], the hedgehog signalling pathway [48], and novel intracellular components of the RIG-I-like receptor (RLR) pathway were all significant for l-lysin [49], whereas the RAC1/PAK1/p38/MMP2 pathway was significant for l-serine and the signalling of hepatocyte growth factor receptor [50,51].

With regard to enzyme inhibitors, curcumin was demonstrated to protect against cholestasis by activating the FXR, which has been identified as a possible therapeutic target for the treatment of cholestasis. Curcumin’s anticholestasis method involved restoring bile acid balance and antagonizing inflammatory responses in an FXR-dependent manner, which resulted in overall cholestasis reduction. It has also been found to play antifibrosis role in the liver [52] and was recently evaluated for its safety and efficacy in fifteen patients with PSC (ClinicalTrials.gov Identifier: NCT02978339), whereas no previous or ongoing studies have evaluated its activity in PBC.

Concerning geldanamycin, an Hsp90 inhibitor, it resulted in a promising drug for the treatment of rheumatoid arthritis because it specifically inhibited the proliferation and inflammation of rheumatoid arthritis fibroblast-like synoviocytes. It has been studied in a variety of haematological and solid malignancy clinical trials.

Interestingly, phenethyl isothiocyanate (PEICT), a PIK3 found in cruciferous vegetables, was found to be enriched. PEICT has been shown to control inflammation by altering the Toll-interleukin-1 receptor domain-containing, adapter-inducing, interferon-dependent signalling pathway of TLRs [53]. No previous clinical trials were found for PEICT in PBC patients. No data concerning the potential use in PBC were found with regard to staurosporine, a cell-permeable alkaloid exhibiting anticancer activity through the PIK3 mechanism.

Another intriguing result is the enrichment found for antioestrogens, along with androgen and integrated breast cancer signalling pathways for LP genes, regardless of their stages. In women and men, the immune system reacts differently. Adult females have higher innate and adaptive immune responses than adult males. Women are more likely than men to develop autoimmune disorders, such as rheumatoid arthritis, multiple sclerosis, autoimmune liver diseases, and PBC, despite having a lower risk of developing most infectious diseases with a higher viral clearance [54].

The significance of oestrogens in autoimmune illnesses has been thoroughly examined, and several lines of evidence and clinical observations indicate that sex hormones play a key role in disease aetiology due to their effects on T cells. Emerging proof, mainly from murine studies, suggests immunosuppressive effects of androgens on T cells [55]. The discovery of alterations in testosterone serum levels in mice connected to the intestinal microbiota should pique interest in the function of the microbiome in sex differences in autoimmune liver disorders, which are linked to an altered intestinal microbiota.

The intestinal immune network for IgA production signalling and IBD pathways was found to be enriched for LP genes—PBC late stages. Antibodies to the endoplasmic reticulum protein calreticulin were found in PBC and autoimmune hepatitis type-1 in a preliminary study [56]. The most striking finding is the high prevalence of IgA anti-calreticulin antibodies and its class pattern in eighty-six PBC patients, suggesting a reactivity of the gut-associated immune system, which could imply that a yet-to-be-identified gut-derived bacterial agent could be a potential actor in the onset of PBC.

The query of whether the classical complement pathway is chronically active in PBC and whether complement activation contributes to the development of bile duct damage remains unanswered. Previously, some authors discovered that complement may not be activated in PBC and that an increase in serum C3 levels is associated with cholestasis [57].

According to a case report, two subjects with PBC had their serum alkaline phosphatase levels drop while taking tamoxifen. Tamoxifen may have this effect by inhibiting cholangiocyte growth and inducing apoptosis via cholangiocyte oestrogen receptors or by activating the pregnane X receptor, which is similar to how UDCA works [58]; this phenomenon warrants further investigation. There are presently no clinical trials testing antioestrogen potential activity in PBC.

Also related to sex-dependent immune response, genistein, an isoflavonoid similar to oestradiol, was also found to be enriched from LP genes, regardless of their disease stage. Amongst its mechanisms, genistein has shown a growth-inhibitory effect on human cholangiocarcinoma cells by reducing AKT and EGFR activation, as well as IL6 production, involving both oestrogen and oestrogen receptors, while also inhibiting inflammatory cell migration. Despite such compounds being found to decrease liver fibrosis and cholestasis induced by a prolonged biliary obstruction in rats [59], genistein has never been studied in a clinical trial setting.

Similar results were found amongst LP genes from unspecified disease stages apart from EGFR-inhibitors. Panitumumab, an IgG2 human monoclonal antibody that inhibits cell proliferation and angiogenesis, significantly reduced the degree of hyperproliferation of the bile duct epithelium and submucosal glands, the collagen fibres of the bile duct wall, the positive rate of EGFR, phosphorylation of mTOR, expression of EGFR, MUC5AC, Ki67, type-I collagen, and G activity [60,61]. Researchers concluded that panitumumab can effectively inhibit the excessive proliferation and stone-forming potential of bile duct mucosa in chronic proliferative cholangitis. Panitumumab was therefore proposed as a promising therapy for the prevention and control of intrahepatic choledocholithiasis caused by chronic proliferative cholangitis. Erlotinib has a similar mechanism, neither drug has been studied in a clinical trial for PBC.

In summary, (i) we conducted a study on a large dataset of PBC curated genes from credible and publicly available sources, proposing potential drug candidates for distinct stages of PBC; and (ii) we obtained a list of new potential disease genes from a novel propagation network algorithm. The latter in silico obtained genes were then enriched for biological pathways and drugs to obtain new potential insights for PBC pathogenesis and treatment.

Although with this study, we identified novel therapeutic targets for prioritization in PBC in an innovative framework that provides a better definition of the PBC molecular landscape, it has several limitations. First, the lack of preclinical and clinical validation of our findings (i.e., newly proposed pathogenic pathways and drug candidates) limited our study, although such a model could be used in future studies for in vitro-directed research. Ideally, this should combine proteomic analyses and functional assays. Nonetheless, an in vitro validation model was beyond the scope of our study.

Second, despite the rigorous bioinformatics methods employed, fibrates did not produce the expected significant results in our analysis, although they are considered effective drugs for treating PBC.

We provided a robust and transparent selection mechanism for prioritizing already approved medicinal or investigational products for repurposing based on recognized unmet medical needs in PBC and sound preliminary data in order to identify research priorities for a better understanding of the mechanisms of action of drug candidates via future ad hoc, in vitro/in vivo tests and clinical trials.

The identification of multiple non-specific liver pathways may shed new light on the extrahepatic pathogenesis of PBC, where gut microbiota, sex hormone-receptor interactions, and bone marrow interplay may all play a role, to varying degrees, at different stages of the disease.

In the first phase, branched-chain amino acids, geldanamycin, tauroursodeoxycholic acid, bioflavonoids (particularly genistein), antioestrogens, curcumin, monoclonal antibodies against osteoclasts (i.e., denosumab), antineovascularisation, and antirheumatic agents are the most interesting therapeutic candidates worthy of evaluation in PBC experiments. Moreover, pharmacological categories such as specific interleukin/EGFR/TNF-alpha inhibitors could be tested in particularly advanced disease stages.

## Figures and Tables

**Figure 1 biomedicines-10-01694-f001:**
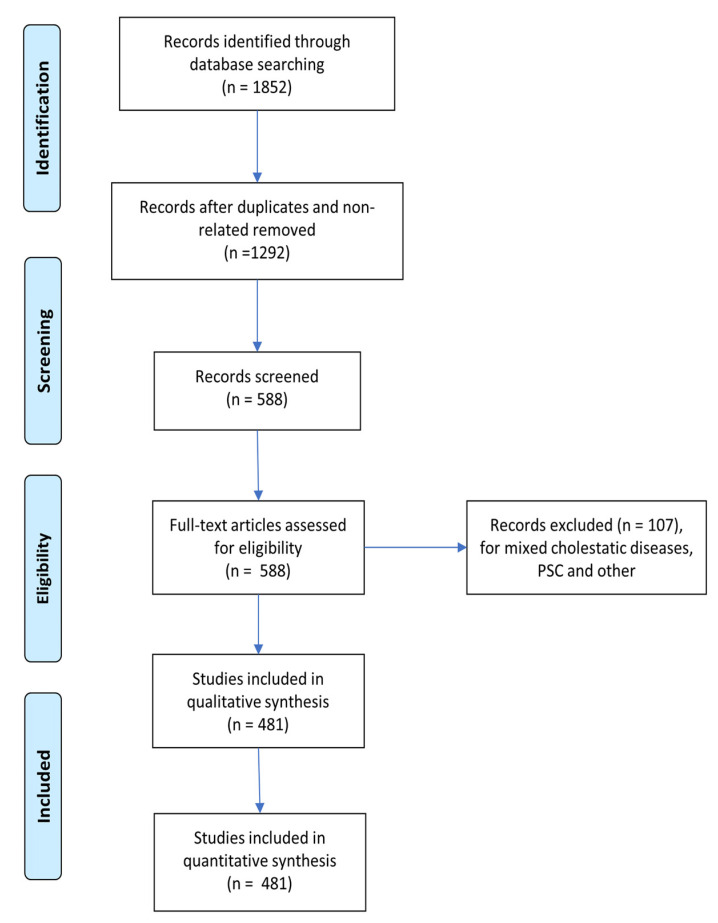
PRISMA flow diagram for study inclusion.

**Figure 2 biomedicines-10-01694-f002:**
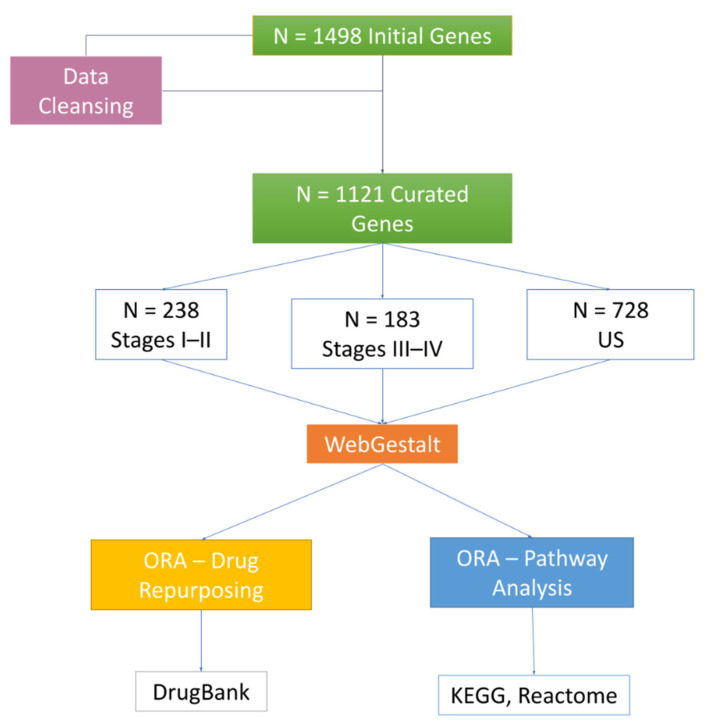
WebGestalt over-representation analysis (ORA) workflow from original seed genes. US: unspecified stages.

**Figure 3 biomedicines-10-01694-f003:**
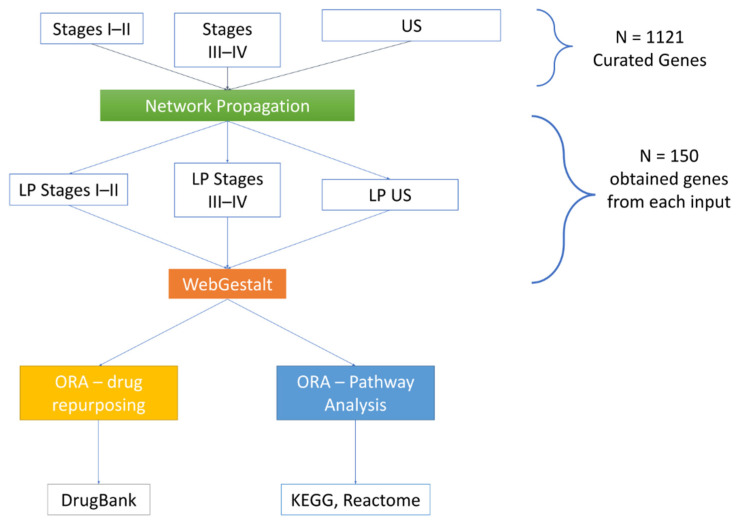
WebGestalt over-representation analysis (ORA) from likely positive (LP) genes obtained after the application of the proposed network-based disease gene prioritisation method. US: unspecified stages.

**Figure 4 biomedicines-10-01694-f004:**
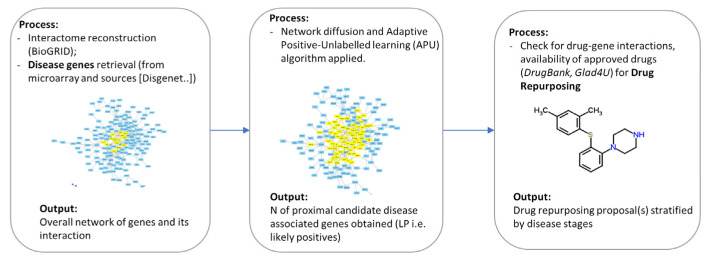
Computational workflow diagram (image adapted from [22]).

**Figure 5 biomedicines-10-01694-f005:**
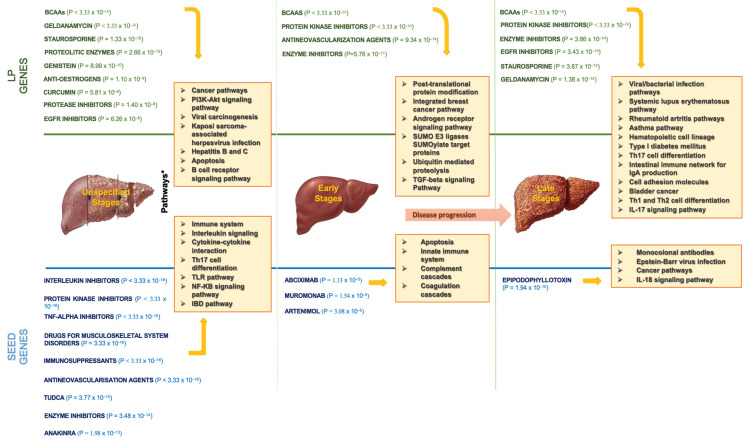
The most enriched drugs and pathways found in LP and seed genes. BCAAs: branched-chain amino acids; EGFR: epidermal growth factor receptor; TNF: tumor necrosis factor; TUDCA: tauroursodeoxycholic acid; TLR: Toll-like receptor; IBD: inflammatory bowel disease.; SUMO: small ubiquitin-like modifier; TGF: trasforming growth factor; IL: interleukin.

## Data Availability

Data is contained within the article and Appendix A.

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
