# Peer review of "Network Proximity-Based Drug Repurposing Strategy for Early and Late Stages of Primary Biliary Cholangitis"

_biomedicines, 2022, doi:10.3390/biomedicines10071694_

Round 1
Reviewer 1 Report
Manuscript ID: biomedicines-1718019.
Manuscript type: Article.
Network Proximity-Based Drug Repurposing Strategy for Early and Late Stages of Primary Biliary Cholangitis
Endrit Shahini , Giuseppe Pasculli, Andrea Mastropietro, Paola Stolfi, Paolo Tieri, Davide Vergni, Raffaele Cozzolongo, Francesco Pesce, Gianluigi Giannelli
Shahini E. et al. have shown in their manuscript a robust bioinformatic network-based strategy to propose possible repurposable drugs for treating primary biliary cholangitis at different stages. For their analysis, the authors selected papers from multiple databases and applied MeSH terms to discriminate and mine targetable genes, after which they applied a network-based putative disease-gene prioritization using BioGRID (both network diffusion process and machine learning algorithm) and the top likely positive genes were then evaluated using WebGestalt. The whole bioinformatic methodology seems accurate, but there are a few comments that should be solved before the acceptance of the manuscript.
a) Major comments
- Although the whole purpose of the work seemed to be the in silico construction of a network that allows the identification of repurposable drugs, the main concern I have is that the authors lack some type of validation of their results. It is expected that the authors select a few of the drugs they proposed, and that they tested them at least in cell-based experiments. They should then analyze the targets that the drugs should attach, maybe by RT-PCR and western blot. This is vital, considering how sporadic this disease is. If the authors could validate their results, then they could provide relevant alternative treatment options for this challenging clinical condition. There are multiple in vivo models the authors could opt to, including:
- a) NOD.c3c4.
- b) Dominant-negative TGF-b receptor II.
- c) IL-2Ra-/-.
All of them spontaneously develop primary biliary cholangitis, and importantly, their biological features resemble those seen in humans.
- An important part of the work was based on the premise of targeting primary biliary cholangitis at different stages, named stages I-II and III-IV, but it was not found in the text a description of such stages. Therefore, it is requested to please add a brief explanation of what these stages are, adding the biological/molecular pathways that are implicated. This could enrich the manuscript by linking the action mechanisms of the possible repurposable drugs with the altered pathways in primary biliary cholangitis.
- An additional outcome of the work done by the authors should be the biological description of the drugs they listed in Table A3. For example, they suggest that for the early stage there could be used L-lysine, L-threonine or L-serine, but they did not propose possible pathways that could be beneficed by treating an autoimmune disease with mere amino acids. Please describe the pathways that could be affected by the proposed treatments.
- Additionally, for geneSets such as PA452621, they described “antineoplastic agents”; for PA164712966, they described “other antineoplastic agents”; and for PA449014, they wrote “cisplatin”. It is confusing that they are not indicating exactly what antineoplastic drug they propose, and why there are “other antineoplastic” drugs. Other than what, than cisplatin? Especially in the case of geneSet PA164712732, the authors only wrote “enzyme inhibitors”. Literally all the drugs inhibit enzymes at some point. The authors must clearly indicate the names of the drugs they proposed, because in the end the main goal of the work is to show repurposing drugs for primary biliary cholangitis.
b) Minor comments
- The way the references are enlisted is confusing, considering that the authors decided to use Roman numerals in the manuscript, but the reference list consists in Arabic numerals. Please ensure consistency in the way the references are listed. If possible, change the Roman numerals for Arabic numerals instead.
- On lines 491 and 495, however, the authors did use Arabic numerals. They referenced works 2, 3 and 61 (line 491). However, reference number 61 is not seen at the end of the list of references (there are only 52 works in that list). Where is reference 61 then?
- According to the references the authors used, it seems like the last reference they list using Roman numerals is xlviii (#48 in Arabic; line 484). If the reference list includes 52 references, there are still missing some works in the text of the manuscript.
- Please rephrase “immune-microenvironment bone marrow/circulating immune cells” (line 67), because the way it is written seems confusing.
- Where the authors referring to “Supplementary Tables A2-A5” (line 209) instead of “Tables A2-A5”? If so, please properly indicate it.
Author Response
REVIEWER n.1
Shahini E. et al. have shown in their manuscript a robust bioinformatic network-based strategy to propose possible repurposable drugs for treating primary biliary cholangitis at different stages. For their analysis, the authors selected papers from multiple databases and applied MeSH terms to discriminate and mine targetable genes, after which they applied a network-based putative disease-gene prioritization using BioGRID (both network diffusion process and machine learning algorithm) and the top likely positive genes were then evaluated using WebGestalt. The whole bioinformatic methodology seems accurate, but there are a few comments that should be solved before the acceptance of the manuscript.
- a) Major comments
- Although the whole purpose of the work seemed to be the in silico construction of a network that allows the identification of repurposable drugs, the main concern I have is that the authors lack some type of validation of their results. It is expected that the authors select a few of the drugs they proposed, and that they tested them at least in cell-based experiments. They should then analyze the targets that the drugs should attach, maybe by RT-PCR and western blot. This is vital, considering how sporadic this disease is. If the authors could validate their results, then they could provide relevant alternative treatment options for this challenging clinical condition. There are multiple in vivo models the authors could opt to, including:
- NOD.c3c4.
- Dominant-negative TGF-b receptor II.
- IL-2Ra-/-.
All of them spontaneously develop primary biliary cholangitis, and importantly, their biological features resemble those seen in humans.
Reply 1a: We thank the reviewer for his comment. The in-vitro validation was not on the target of this current study. Nonetheless, we are planning a separate work focused on the in-vitro validation of the most promising drugs with ad-hoc experiments. Given that this additional task involves a considerable investment of resources and time, we think however that the manuscript as it is now is timely and needed and could make a significant contribution to the field of PBC studies.
- An important part of the work was based on the premise of targeting primary biliary cholangitis at different stages, named stages I-II and III-IV, but it was not found in the text a description of such stages. Therefore, it is requested to please add a brief explanation of what these stages are, adding the biological/molecular pathways that are implicated. This could enrich the manuscript by linking the action mechanisms of the possible repurposable drugs with the altered pathways in primary biliary cholangitis.
Reply 1b: We thank the reviewer for his comment. We have now added a better explanation, supported by a proper reference (Scheuer P. Primary biliary cirrhosis. Proc R Soc Med 1967;
60:1257–1260) of the PBC different stages. We deemed it not useful to lengthen the description of such histological stages, which are commonly used in the field, in order to avoid exceeding the paper word count limits. With regards of the disease stages associated pathways, we already presented such results in Supplementary Table A4.
- An additional outcome of the work done by the authors should be the biological description of the drugs they listed in Table A3. For example, they suggest that for the early stage there could be used L-lysine, L-threonine or L-serine, but they did not propose possible pathways that could be beneficed by treating an autoimmune disease with mere amino acids. Please describe the pathways that could be affected by the proposed treatments.
Reply 1c: ​​We thank the reviewer for his comment. A brief assessment of the pathways interacting with the genes that were found to interact with such BCAAs (with their potential implications in PBC) has now been implemented within the discussion section of the main manuscript. Such results were also added within the Supplementary Files 2, 3 and 4. As previously stated, we have postulated distinct BCAA-related pathways.
To respond more directly to the reviewer's statement about the utility of simple amino acids in the treatment of a complex immune-mediated disease like PBC, the analysis of how BCAAs activate different pathways demonstrates how these amino acids work on multiple targets through multiple mechanisms, including ERα, which has been found to be abnormally up-regulated in PBC patients (Cao H, et al ). Signal transducer and activator of transcription 3 (STAT3) phosphorylation in human intrahepatic biliary epithelial cells has been shown to stimulate ER-mediated production of pro-inflammatory cytokines (Cao H, et al.). STAT3 has been also identified as the major regulator of mammary gland involution (Kilanczyk E et al.). BCAAs may also target the Hedgehog system, which regulates liver tissue morphogenesis and has been linked to the response to biliary injury in PBC (Jung Y et al.). Finally, BCAAs may help restore the equilibrium of matrix metalloproteinases that has been altered in experimental biliary fibrosis models (Ghaffari K et al, and Kossakowska AE et al.).
- Additionally, for geneSets such as PA452621, they described “antineoplastic agents”; for PA164712966, they described “other antineoplastic agents”; and for PA449014, they wrote “cisplatin”. It is confusing that they are not indicating exactly what antineoplastic drug they propose, and why there are “other antineoplastic” drugs. Other than what, than cisplatin? Especially in the case of geneSet PA164712732, the authors only wrote “enzyme inhibitors”. Literally all the drugs inhibit enzymes at some point. The authors must clearly indicate the names of the drugs they proposed, because in the end the main goal of the work is to show repurposing drugs for primary biliary cholangitis.
Reply 1d: We thank the reviewer for his comment. We have now expanded the analyses for only the PA452621 and PA164712966 gene subsets in order to retrieve more precise information on the potential drugs interacting with such disease genes. The results of the analysis are in Supplementary Material (Supplementary Files 5, 6 and 7). Specifically, as for “antineoplastic agents” (PA452621), the most relevant drugs were vincristine, vinblastine, epipodophyllotoxin, topoisomerase I inhibitors, daunorubicin, etoposide, fluorouracil, platinum compounds, cisplatin. Also, as for “other antineoplastic agents” (PA164712966) neratinib, panitumumab, leuprolide, nilutamide, XL765, afatinib, selumetinib, crizotinib, anastrozole, amuvatinib, ibrutinib, alemtuzumab, trabectedin, erlotinib, methoxsalen, nutlin-3, anti-androgens, docetaxel, epipodophyllotoxin, anti-estrogens, platinum, sorafenib, camptothecin, taxanes, etoposide, cisplatin, and protein kinase inhibitors were the most significant. Additionally, as for “enzyme inhibitors” (PA164712732) purvalanol, staurosporine, salvianolic acid b, berberine, resveratrol, curcumin, methoxsalen, acetylcysteine, genistein, and trichostatin A were the most significant.
- b) Minor comments
- The way the references are enlisted is confusing, considering that the authors decided to use Roman numerals in the manuscript, but the reference list consists in Arabic numerals. Please ensure consistency in the way the references are listed. If possible, change the Roman numerals for Arabic numerals instead.
Reply 1e: ​​Thanks to the reviewer for pointing this out. We have accordingly modified the references list, as suggested by you, from Roman numerals into Arabic ones throughout the whole text.
- On lines 491 and 495, however, the authors did use Arabic numerals. They referenced works 2, 3 and 61 (line 491). However, reference number 61 is not seen at the end of the list of references (there are only 52 works in that list). Where is reference 61 then?
Reply 1f: ​​Thanks to the reviewer for pointing this out. We have accordingly modified the references list, as suggested by you, from Roman numerals into Arabic ones (lines 491 and 495). We also added the reference number 61 as it went missing probably throughout the rendering process after submission.
- According to the references the authors used, it seems like the last reference they list using Roman numerals is xlviii (#48 in Arabic; line 484). If the reference list includes 52 references, there are still missing some works in the text of the manuscript.
Reply 1g: ​​Thanks to the reviewer for pointing this out. We have accordingly modified the references list.
- Please rephrase “immune-microenvironment bone marrow/circulating immune cells” (line 67), because the way it is written seems confusing.
Reply 1h: ​​We have accordingly rephrased “immune-microenvironment bone marrow/circulating immune cells” (line 67) into “bone marrow microenvironment”.
- Where the authors referring to “Supplementary Tables A2-A5” (line 209) instead of “Tables A2-A5”? If so, please properly indicate it.
Reply 1i: We thank the reviewer for this comment. We have now corrected the table referral with the correct one.
Reviewer 2 Report
biomedicines-1718019, Network Proximity-Based Drug Repurposing Strategy for Early and Late Stages of Primary Biliary Cholangitis
The manuscript presents an interesting research based on the correlation of genes, diseases and drugs in order to find therapeutically solutions for primary biliary cholangitis. The manuscript has value and could be useful for journal’s readers after some corrections and improvement.
A major problem of this study and also of similar ones is that the pathways are interconnected and a disease has multiple causes, and not a single one. The relationships between causes can be synergistic or antagonistic and many co-factors can appear. Also, a drug can interact in multiple points with various pathways and the final effect is not always easy to predict. The results of the manuscript (row 351) “the immune system, cancer pathways, interleukin signaling, cytokine-cytokine receptor interaction, Th17 cell differentiation, and toll-like receptor (TLR)/NF-kB signaling/IBD pathways musculoskeletal system disorders, TUDCA, immunosuppressants, antirheumatic agents” clearly reflect this problem. All the described pathways have in common the interaction with the immune system. And this is nothing new. In my opinion, the results add little new information.
The authors should be more critical with their results considering the problems associated with the methods used.
The editing should be corrected to fit the journal’s requirements. See for example the numbering of references. There are many other style problems that need corrections. The authors should use the equation module of OfficeWord to draw all the equations presented in the supplementary section. The equations should be numbered.
On row 125, the authors should present the mathematical formula for the GDA score.
The whole section 143 to 154 has no references. The proper references should be provided. The same problem was observed at row 177.
Row 217, it should be 3.77E-15, and not 3.77e-15. Please check all the manuscript.
Figure 5 should be rotated so it can be read better.
On row 350 the authors declare “statins were among the potential drug classes”, but their results include only simvastatin and atorvastatin. Please check and correct.
In the supplementary section present what PMID represents (table A1). In table A2 is a “skeleton” label. Please check. Overall, the labels in the “description” columns loo strange from a pharmacology point of view. Some are very large, like ”carbohydrates”, “vaccines” or “respiratory system”, and some look wrong, like “copper” or “podofilox”. Podofilox is the name of a commercial product, not a substance. It should be Podophyllotoxin. Similar problems can be found in table A3. “Enzyme inhibitors” it is such a vast category and really confusing. Also, EGFR inhibitors can be described as “Protein kinase inhibitors”, “antineoplastic agents”, “Enzyme inhibitors”. In my opinion this overlapping of categories is confusing and should be corrected.
In table A3, see “nitrile”, “amide”, or “ethers” are also very large categories for a chemical point of view. They can also overlap. A drug can have both an amide group and a nitrile one. In my view it adds little information presenting data on large categories.
Author Response
REVIEWER n.2
The manuscript presents an interesting research based on the correlation of genes, diseases and drugs in order to find therapeutically solutions for primary biliary cholangitis. The manuscript has value and could be useful for journal’s readers after some corrections and improvement.
- A major problem of this study and also of similar ones is that the pathways are interconnected and a disease has multiple causes, and not a single one. The relationships between causes can be synergistic or antagonistic and many co-factors can appear. Also, a drug can interact in multiple points with various pathways and the final effect is not always easy to predict. The results of the manuscript (row 351) “the immune system, cancer pathways, interleukin signaling, cytokine-cytokine receptor interaction, Th17 cell differentiation, and toll-like receptor (TLR)/NF-kB signaling/IBD pathways musculoskeletal system disorders, TUDCA, immunosuppressants, antirheumatic agents” clearly reflect this problem. All the described pathways have in common the interaction with the immune system. And this is nothing new. In my opinion, the results add little new information. The authors should be more critical with their results considering the problems associated with the methods used.
Reply 2a: ​​We thank the reviewer for his right comments. As already stressed in the study limitations, the current study still lacks an in-vitro validation of these preliminary findings which would have provided a much deeper understanding of the potential activity of the proposed drugs towards the many various PBC pathways. Nonetheless, as already mentioned above, we are currently planning to undertake such tests for a separate, more in-vitro/wet-lab-based research paper after this first introductory/methodological one by studying the most innovative and significant drugs with no previous clinical data available at the moment.
- The editing should be corrected to fit the journal’s requirements. See for example the numbering of references. There are many other style problems that need corrections. The authors should use the equation module of OfficeWord to draw all the equations presented in the supplementary section. The equations should be numbered.
Reply 2b: We thank the reviewer for his comment. We have corrected the equation rendering (plus adding the numbering) and the references that were probably out of sync throughout the submission process. The new methodology file is Supplementary File 1.
- On row 125, the authors should present the mathematical formula for the GDA score.
Reply 2b: As suggested, we have added the mathematical formula for the GDA score.
- The whole section 143 to 154 has no references. The proper references should be provided. The same problem was observed at row 177.
Reply 2c: As suggested, we have added a specific reference to the section from 143 to 154.
- Row 217, it should be 3.77E-15, and not 3.77e-15. Please check all the manuscript.
Reply 2d: As suggested, we have changed 3.77E-15 into 3.77e-15 (row 217).
- Figure 5 should be rotated so it can be read better.
Reply 2e: Unfortunately, figure 5 can’t be rotated as it overflows the paper horizontal limits. Maybe, the Editorial Office can resolve this issue in the production phase after your suggestion.
- On row 350 the authors declare “statins were among the potential drug classes”, but their results include only simvastatin and atorvastatin. Please check and correct.
Reply 2f: We modified the sentence “statins were among the potential drug classes” into “simvastatin and atorvastatin were among the potential drug classes”.
- In the supplementary section present what PMID represents (table A1). In table A2 is a “skeleton” label. Please check. Overall, the labels in the “description” columns look strange from a pharmacology point of view. Some are very large, like “carbohydrates”, “vaccines” or “respiratory system”, and some look wrong, like “copper” or “podofilox”. Podofilox is the name of a commercial product, not a substance. It should be Podophyllotoxin. Similar problems can be found in table A3. “Enzyme inhibitors” it is such a vast category and really confusing. Also, EGFR inhibitors can be described as “Protein kinase inhibitors”, “antineoplastic agents”, “Enzyme inhibitors”. In my opinion this overlapping of categories is confusing and should be corrected.
Reply 2g: We thank the reviewer for his comment and, accordingly, specified what PMID represents in the supplementary section (PubMed unique identifier). Unfortunately, Webgestalt results also tend to generate wide categories as results. As for reviewer 1 comment as well, whenever possible, we exploded the results from one of those broader categories (e.g. Antineoplastic agents) in a further Webgestalt analysis providing for more detailed compounds. Nonetheless, this could not be guaranteed for every of such broad drug categories obtained as it might have provided some less reliable results in terms of drug proposals. However, the main and most significant results we obtained (and commented on within the main text) were present as single chemical entities/drugs from their Webgestalt analysis.
We changed Podofilox into podophyllotoxin, as suggested. With regards to the tables, we've listed the categories in the same order as they appeared in the Webgestalt analyses' results, while in the discussion, we've listed the repurposed drugs we deemed to be most important from a clinical point of view. Furthermore, we changed and homogenized the overlapping categories.
- In table A3, see “nitrile”, “amide”, or “ethers” are also very large categories for a chemical point of view. They can also overlap. A drug can have both an amide group and a nitrile one. In my view it adds little information presenting data on large categories.
Reply 2h: We thank the reviewer for his comment and link the reply to the previous answer adding that in order to discern between similar/overlapping classes another criterion (e.g. p-value, FDR and Enrichment Ratio) could be taken into account (i.e. every result coming from the Webgestalt analysis must be first evaluated in terms of results robustness and then critically evaluated from a clinical point of view before being deemed as a potential rational candidate for PBC treatment; hence our discussion section just focusing on the most interesting compounds satisfying such criteria).
Round 2
Reviewer 1 Report
Although the authors have adequately answered the comments, they are still lacking the validation of their results. They said that the validation is supposed to be done in a subsequent work, but I recommend to do it in this one. Otherwise, their results could sound speculative.
Author Response
We certainly agree that ideally, confirmatory experiments would provide additional insights. However, we might argue that in this case it can be considered a non/essential requirement for the approval of our accurate work. Indeed, we planned to conduct a speculative in-silico study following the theme of this journal's Special issue titled "Bioinformatics and its Application in Biomedicine", and we provided meaningful contributions to the prioritization of novel therapeutic targets in PBC in an innovative framework that offers a better definition of the PBC molecular landscape (i.e., newly proposed pathogenic pathways and drug candidates). Accordingly, the in vitro validation model is beyond the scope of this paper. The Guest editor of the special issue is not part of the decision process due to a potential conflict of interest with one of the co-authors, however, he agrees. Moreover, in the most favorable scenario, our study will be made experimental, after a long time due to the elevated estimated expenses and the necessity for additional funds. Also, we envisage this follow-up study as a combination of proteomic analyses and functional assays, which we believe would be outside this paper's aim. As a result, we have added a paragraph in the last part of the discussion section (page 14) in response to your objection to completeness.
Reviewer 2 Report
The authors made some changes on their paper and improved its quality. The articles still needs some corrections. The authors should prepare a small conclusion section in which to clearly present their results. They should clearly and sincerely present what does this research really adds new. The authors should have a more objective style when presenting their results.
Author Response
We thank the reviewer for his useful suggestion. Accordingly, we have organized a short conclusion section to explain our principal results.
“...The identification of multiple non-specific liver pathways may shed new light on the extrahepatic pathogenesis of PBC, where gut microbiota, sex hormone-receptor interactions, and bone marrow interplay may all play a role to varying degrees at different stages of the disease. In the first phase, branched-chain amino acids, geldanamycin, tauroursodeoxycholic acid, bioflavonoids (particularly genistein), anti-oestrogens, curcumin, monoclonal antibodies against osteoclasts (i.e., denosumab), antineovascularisation, and antirheumatic agents would be the most interesting therapeutic candidate’s worthy of evaluation in the PBC experiments. Moreover, pharmacological categories such as specific interleukin/EGFR/TNF-alpha inhibitors could be tested in particularly advanced disease stages”.
Round 3
Reviewer 1 Report
The authors are in agreement that the experimental part of the manuscript is still lacking. Being part of a special issue that already contains bioinformatic-based papers with validated results in vitro, in vivo and even using actual patient samples (i.e., 10.3390/biomedicines10030530, 10.3390/biomedicines10020402, 10.3390/biomedicines9121817, 10.3390/biomedicines9101314, 10.3390/biomedicines9091120, 10.3390/biomedicines9040410), there is no real explanation that sustains the missing validation results. The subject of research is of high relevance and must be replicated. The authors claim that they visualize the experimental part as a separate paper, but if they already plan to have such results, they must add them here. Additionally, the authors point out a possible conflict of interest with the guest editor, and thus, if the manuscript gets accepted in its actual status without any form of validation, then the same guest editor would show a scientific misconduct by favoring incomplete research papers. MDPI is not a predatory publisher, and therefore all manuscripts that are accepted for publication require to follow a proper scientific methodology.